# Short Chain Fatty Acid Biosynthesis in Microalgae *Synechococcus* sp. PCC 7942

**DOI:** 10.3390/md17050255

**Published:** 2019-04-28

**Authors:** Yi Gong, Xiaoling Miao

**Affiliations:** 1State Key Laboratory of Microbial Metabolism, School of Life Sciences & Biotechnology, Shanghai Jiao Tong University, 800 Dongchuan Road, Shanghai 200240, China; gongvsyi@126.com; 2Joint International Research Laboratory of Metabolic & Developmental Sciences, Shanghai Jiao Tong University, Shanghai 200240, China; 3Biomass Energy Research Center, Shanghai Jiao Tong University, Shanghai 200240, China

**Keywords:** microalgae, *Synechococcus* sp. PCC 7942, short chain fatty acids, β-ketoacyl ACP Synthase

## Abstract

Short chain fatty acids (SCFAs) are valued as a functional material in cosmetics. Cyanobacteria can accumulate SCFAs under some conditions, the related mechanism is unclear. Two potential genes Synpcc7942_0537 (*fabB/F*) and Synpcc7942_1455 (*fabH*) in *Synechococcus* sp. PCC 7942 have homology with *fabB/F* and *fabH* encoding β-ketoacyl ACP synthases (I/II/III) in plants. Therefore, effects of culture time and cerulenin on SCFAs accumulation, expression levels and functions of these two potential genes were studied. The results showed *Synechococcus* sp. PCC 7942 accumulated high SCFAs (C12 + C14) in early growth stage (day 4) and at 7.5g/L cerulenin concentration, reaching to 2.44% and 2.84% of the total fatty acids respectively, where *fabB/F* expression was down-regulated. Fatty acid composition analysis showed C14 increased by 65.19% and 130% respectively, when *fabB/F* and *fabH* were antisense expressed. C14 increased by 10.79% (*fab(B/F)*^−^) and 6.47% (*fabH*^−^) under mutation conditions, while C8 increased by six times in *fab(B/F)*^−^ mutant strain. These results suggested *fabB/F* is involved in fatty acid elongation (C <18) and the elongation of *cis*-16:1 to *cis*-18:1 fatty acid in *Synechococcus* sp. PCC 7942, while *fabH* was involved in elongation of fatty acid synthesis, which were further confirmed in complementary experiments of *E. coli*. The research could provide the scientific basis for the breeding of SCFA-rich microalgae species.

## 1. Introduction

Short chain fatty acids (SCFAs) are usually defined as the carbon number of fatty acids between 6 and 14 [1]. Due to corresponding glycerides with extraordinary characteristics such as low viscosity, high extensibility, low freezing point, low surface tension, high transparency and oxidation stability, SCFAs have very broad application in cosmeceuticals, nutraceuticals, nutritional supplements, chemical industry, etc. [2]. Currently, main sources of SCFAs are tropical plants, such as coconut and palm trees [3,4]. However, people have to turn their attention to the potential oleaginous microorganisms because of the high cost of vegetable oils and the limitation of climate and land resources [5]. It is widely reported that bacteria, mold, yeast and microalgae are important oleaginous microorganisms [5], of which microalgae has received much attention for its unique advantages such as fast growth rate, high oil content, high photosynthetic efficiency, low land requirements and environmental protection [6].

There are abundant algae in the ocean, the unique ecological environment of which causes the special bioactive metabolites accumulation in the algae. These metabolites have a series of biological effects, such as moisturizing [7], bacteriostasis [8], anti-inflammatory [9], antivirus [10], inhibiting the growth of tumor cells [11], and resisting ultraviolet radiation [12]. It was found that a kind of ancient cyanobacteria has antioxidant, immune and other biological activities [13]. In recent years, safety problems are often exposed in the cosmetics industry, when algae bioactive substances could be used as raw materials for new cosmeceuticals due to their low toxicity and high safety. Therefore, algae would be increasingly valued as a functional material in the field of cosmetics. At present, research on algae focuses on the active substances such as minerals, bioactive peptides, natural pigments, enzymes, polysaccharides and unsaturated fatty acids, etc. [14], while little research are about short carbon chain fatty acids. It was found that short carbon chain fatty acids were widely used in cosmetics, where they could replace white oil, lanolin and squalane. Compared with squalane, short carbon chain fatty acids are more easily absorbed by the skin and can be rapidly oxidized and metabolized. In addition, short carbon chain fatty acids have emulsifying stability and antioxidant properties, which can make cosmetics more uniform and delicate, improve product quality and storage period. In suntan lotion, short carbon chain fatty acids are non-greasy and uncomfortable feeling. In lipsticks, the short carbon chain fatty acids can eliminate the unique smell of lanolin, making the matrix tissue delicate, the pigment dispersion uniform and improving the surface gloss and spread ability [15].

Microalgae, as a high-quality oleaginous aquatic microorganism, are primarily used for biodiesel, polyunsaturated fatty acids and pigments, but rarely for SCFAs. The content of SCFAs in microalgae was not high under normal growth conditions [16,17,18,19,20,21]. Some microalgae can accumulate SCFAs when cultivated in stressed conditions [22], but the related mechanism of this is unclear. Studies on the synthesis mechanisms of SCFAs are mostly conducted in plants and bacteria [4], and mainly focused on the key enzymes such as thioesterases (TEs) and β-ketoacyl ACP (Acyl Carrier Protein) synthases (KAS). TEs are mainly present in plants and some eukaryotic microalgae, associated with the termination of the fatty acid synthesis cycle by catalyzing ACPs to remove from acyl-ACPs, producing free fatty acids [23,24]. Past efforts to increase medium chain fatty acid (MCFA) production in microalgae by genetic modifications of chain-length specific TEs have met with limited success, because of high specifics and substrate preferences of TEs. β-ketoacyl ACP synthases are associated with the carbon chain elongation of fatty acids, which are classified as KAS I, KAS II and KAS III, encoded by *fabB*, *fabF* and *fabH*, respectively [25,26]. It is reported that KAS III could catalyze the combination of acetyl-CoA with malonyl-CoA to generate 4:0-ACP, KAS I could catalyze 4:0-ACP to generate 16:0-ACP and KAS II could catalyze 16:0-ACP to generate 18:0-ACP as well as control the ratio of 16:0-ACP/18:0-ACP [27]. Verwoert et al. [28] overexpressed the *KAS III* gene from *E. coli* in rapeseed, causing changes in fatty acid composition, which decreased the content of C18:1 and increased the content of C18:2 and C18:3. Dehesh et al. [29] overexpressed the *KAS III* gene from spinach in three kinds of plants (Tobacco, *Arabidopsis thaliana*, Rape), leading to C16:0 fatty acid accumulation. Research on cyanobacteria *Synechococcus* sp. PCC 7002 revealed that KAS III initialized the synthesis of fatty acids circle, causing the condensation of malonyl ACP and acetyl CoA to form acetyl–acetyl ACP, which was thought of the only rate-limiting step in fatty acid synthesis of cyanobacteria [30]. The result is different from that of *E. coli*, where KAS III was not rate-limiting.

Although the fatty acids composition of microalgae varied with species and strains [31], but some cyanobacteria had strong ability to synthesize short carbon chain fatty acids. In the past, filamentous cyanobacteria *Trichodesmium erythraeum* has been found to produce 27% to 50% C10 fatty acids [32]. Later, Karatay et al. [33] reported *Synechococcus* sp. could accumulate about 23.8% of caprylic acid (C10:0) and myristic acid (C14:0) under nitrogen deficiency condition. Therefore, cyanobacteria may be an ideal species for the production of short carbon fatty acids. In addition, cyanobacteria, as prokaryotes, are more suitable for gene manipulation than eukaryotic microalgae, and some of which have completed whole genome sequencing. The sequencing information could also provide guidance for our research. Thus, *Synechococcus* sp. PCC 7942, whose genome have been sequenced, was used in the present study to explore the effects of cerulenin and culture time on the synthesis of SCFAs, the functions and expression regulation of genes related to the SCFAs synthesis were further investigated. The results may provide scientific guidance for the future development of short chain fatty acids resources in microalgae by means of metabolic engineering and molecular biology.

## 2. Results and Discussion

### 2.1. Fatty Acid Compositions of Synechococcus sp. PCC7942 under Different Conditions

Microalgae synthesize and store different types of lipids in a single cell [32]. In contrast to the conventional fatty acids (FAs) composition of microalgae that contain long chain fatty acids (LCFAs) from C16 to C18, SCFAs (from C8 to C14) are generally classified to be unusual. Generally, the content of SCFAs in microalgae is relatively low under normal growth conditions. In order to better understand the synthesis mechanism of SCFAs in microalgae from a physiological perspective preliminarily, the fatty acid profiles of *Synechococcus* sp. PCC 7942 under different culture time periods was investigated.

Table 1 shows the dynamic variations of fatty acids during the whole culture process. As shown in Table 1, the main fatty acid was C16 under different culture time periods, accounting for about 70%–80% of the total fatty acids, while little contents of LCFAs from C20 to C22 were detected. At the early growth stage (2 days–6 days), high content of SCFAs (C14 + C12) were observed, reaching highest content (2.44%) at day 4, among which C14:1 was 1.70% and C14:0 was 0.74% (Table 1). After day 8, contents of C14:0 and C14:1 did not change significantly, which were in the range of 0.3%–0.6% and 0.5%–0.9%, respectively, while C12 fatty acid could not be detected. These results suggested that the accumulation of SCFAs mainly occurred in the early growth stage (2 days–6 days), with C14 and C12 as the main composition. To our knowledge, this was the first report of SCFAs accumulation in the early growth stage in cyanobacteria.

The fatty acid synthesis system of plants and bacteria belonged to the type II, where the KASs were related to the fatty acid carbon chain elongation and generally divided into KAS I (FabB), KAS II (FabF) and KAS III (FabH) [25,26]. It was reported that fatty acid synthase inhibitors have effects on the fatty acids carbon chain elongation in bacteria [34,35], among which cerulenin mainly played its role on KAS I and KAS II. In order to further understand the SCFAs synthesis in *Synechococcus* sp. PCC 7942, effects of different cerulenin concentrations on SCFAs accumulation were studied.

As shown in Table 2, the content of SCFAs (C12 + C14) accumulated as high as 2.84% of the total fatty acids under cerulenin concentration of 7.5 g/L, among which the content of C14 fatty acids increased from 1.58% (0 g/L) to 2.75% (7.5 g/L), increasing by 74.05% (*p* < 0.05), and C12 fatty acids increased from 0.04% to 0.09%, increasing by 125% (*p* < 0.05). At this time, C16 fatty acids decreased from 63.49% to 57.35%, decreasing by 9.67%, C18 fatty acids decreased from 18.44% to 13.7%, decreasing by 25.7%, respectively. Israel et al. [36] found that after 40 min of adding cerulenin into the medium with *E. coli*, 90% of oil synthesis and 25% of RNA and DNA synthesis were inhibited, but the protein synthesis was not affected. In addition, a previous study [37,38] on antibacterial drug targets showed that cerulenin has specific inhibition on KAS I and KAS II, which are responsible for the carbon chain elongation of C4–C16 fatty acids in bacteria. As shown in Table 2, the contents of longer chain fatty acids (C18 + C16) decreased and the contents of SCFAs (C12 + C14) increased, with the increase of cerulenin concentration. These results suggested that KAS I/II in *Synechococcus* sp. PCC 7942 might be inhibited, which resulted in the increase of SCFAs (C12 + C14).

Based on the above results, it is assumed that the low expression levels of the related genes *fabB/F* (KAS I/II) in *Synechococcus* sp. PCC 7942 might inhibit the carbon chain elongation of fatty acids, leading to the sufficient accumulation of SCFAs at this stage. Therefore, the related genes expression levels under the above conditions were further investigated.

### 2.2. Genes Expression under Different Conditions in Synechococcus sp. PCC7942

Bioinformatic analysis showed that Synpcc7942_0537 and Synpcc7942_1455 in *Synechococcus* sp. PCC 7942 might have homology with *fabB/F* and *fabH* of plants and bacteria.

As mentioned above, the influences of culture time period and cerulenin concentration on the fatty acid compositions of microalgae were significant. In order to understand the relationship between genes expression level and fatty acids (FAs) carbon chain elongation, the influences of cerulenin concentration and culture time on transcriptional levels of the relative genes (*fabB/F* and *fabH*) were explored.

As shown in Figure 1a, the expression level of *fabB/F* was relatively low in the early growth stage (2 days–6 days), and reached the highest expression level at day 8, then decreased in the later period of culture time (10 days–16 days). The overall trend of expression level of *fabB/F* increased first and then decreased, which was consistent with the variations of fatty acid composition (Table 1), that is to say, the SCFAs mainly accumulated in early growth stage. The result indicated that *fabB/F* played an important role on the synthesis of SCFAs. On the other side, during the whole cultivation phase, the expression level of *fabH* was not high enough, and showed a trend of gradual decrease, only with relatively high expression level in the early growth stage (2 days–6 days) (Figure 1b). Inhibition of the expression of *fabB/F* will hinder the transformation of SCFAs (C12, C14) to relatively longer chain fatty acids (C16, C18), leading to SCFAs accumulation, which also verified the previous hypothesis that *fabB/F* is a key gene promoting the synthesis of SCFAs (C12, C14) to longer chain fatty acids (C16, C18).

As shown in Figure 1c, under cerulenin concentration of 7.5 g/L, the expression level of *fabB/F* was relatively high at the early growth stage (2 days–8 days), then decreased significantly with the extension of culture time and reached the lowest expression level in the late growth stage (12 days–16 days). These results suggested that cerulenin could change the expression levels of *fabB/F* in *Synechococcus* sp. PCC 7942. Ter Beek et al. [39] also found that cerulenin could change the expression levels of fab (fatty acid biosynthesis) genes in fatty acid synthesis of *Bacillus subtilis*. The dynamic trend of *fabB/F* expression at 7.5 g/L of cerulenin was consistent with variation of SCFAs composition, compared with that in the absence of cerulenin or at low concentration of cerulenin (Table 2). The expression level of *fabH* gene did not change significantly during the whole cultivation period under 7.5 g/L cerulenin concentration (Figure 1d), which indicated that cerulenin also has specific inhibition on KAS I and KAS II in *Synechococcus* sp. PCC 7942. However, high cerulenin concentration could cause the inhibition of the growth of *Synechococcus* sp. PCC 7942. When the cerulenin concentrations were between 0 g/L and 7.5 g/L, the OD_730_ value of the same culture time period decreased with the increase of the concentration of cerulenin. At 7.5 g/L cerulenin concentration, the OD_730_ value was the lowest, which was about one third of that at 0 g/L (Appendix A). Therefore, although there was no significant change during the whole cultivation period, the *fabH* expression level at 7.5 g/L cerulenin was much lower (Figure 1d) compared with that in the absence of cerulenin (Figure 1b). This is probably caused by the growth inhibition at 7.5 g/L cerulenin concentration (Appendix A). So, the above dynamic expressions of two related genes in *Synechococcus* sp. PCC 7942 may be the result of both growth and gene inhibition.

### 2.3. Fatty Acid Compositions in fabB/F and fabH Sense/Antisense Expression Strains

It is reported that deletion of some important enzymes in the fatty acid synthesis pathway may lead to cell death [40]. Therefore, sense and antisense expression strains of *fabB/F* and *fabH* were constructed first to investigate the functions of the two genes in the process of fatty acids synthesis. Chen et al. [41] reported that inhibiting the expression of phosphoenol pyruvate carboxylase (PEPC) gene in rapeseed by antisense RNA led the oil content of transgenic rapeseed to 6.4%–18%, which was higher than that of the control group. Song et al. [42] successfully antisense expressed the encoding gene (*pcc*) of PEPC in *Synechococcus* sp. PCC 7002, resulting in an oil content increase. It can be seen that antisense expression is very feasible in improving the oil content of microalgae.

As shown in Table 3, the SCFAs contents were not significantly changed in *Synechococcus* sp. PCC 7942 harboring sense plasmids pRL-Sense-*fabB/F* (Synpccw7942_0537) and pRL-Sense-*fabH* (Synpccw7942_1455) (The sense and antisense plasmids were all derived from pRL 489). Research in plants showed *fabB/F* was responsible for the longer chain fatty acids elongation, while *fabH* was related to the initiation of fatty acids synthesis and extending C2 to C4 [27]. The present study suggested that acyl-ACPs for longer chain fatty acids synthesis may enter the next step of fatty acid synthesis, which could not result in a massive accumulation of SCFAs (Table 3). It was also found that overexpression of KAS IV in a MCFA-producing strain of *Dunaliella tertiolecta* could allow MCFA accumulation [43], which lies in the control of a KAS enzyme to generate MCFA acyl-ACP substrates. We believed that enlargement of the MCFA acyl-ACP substrate pool by overexpressing KAS IV and concentration of MCFA acyl-ACP substrate pool caused by KAS enzyme all could lead to MCFAs accumulation [24]. Although the overexpression of *KAS III* gene in *E. coli* could change the fatty acid profile of *E. coli*, when C14:0 increased and C18:1 decreased [26], which were different from our results of *Synechococcus* sp. PCC 7942 with pRL-Sense-*fabH* (Table 3), indicating that the Synpccw7942_1455 (*KAS III*) gene had species difference. Furthermore, eukaryotic microalgae (*Dunaliella tertiolecta*) and bacteria/cyanobacteria are innately different, given that eukaryotic microalgae are capable to store free FAs in the form of SCFAs.

The content of C18 of *Synechococcus* sp. PCC 7942 containing pRL-Antisense-*fabB/F* decreased from 18.44% of wild-type to 2.15% of antisense expression strain (Table 3), with a decrease of 88.34%. Meanwhile, the content of C14 in *Synechococcus* sp. PCC7942 containing pRL-Anti-*fabB/F* increased from 1.58% of the wild type to 2.61% of antisense expression strain (Table 3), with an increase of 65.19% (*p* < 0.05). This is because that the antisense gene fragment could inhibit the expression level of *fabB/F*. The expression levels of *fabB/F* in antisense expression of *fabB/F* strains were down-regulated (*p* < 0.05) as compared with that in the wild type strain during the whole growth period (Appendix A). In addition, as shown in Table 3, C16:2 in *fabB/F* antisense expression strain decreased from 0.27% to 0.13%, C18:2 decreased from 0.81% to 0.18%, C17 (17:0 and 17:1) fatty acids decreased from 1.3% to 0.87%. By referring to the studies of other microorganisms [3], *fabB/F* might be related to the extension of longer chain fatty acids. Further analysis showed that C16:1 of the antisense expression strain increased from 31.75% to 37.20% and C18:1 decreased from 15.84% to 1.25% (Table 3), which was consistent with the function of *fabF* for extending *cis*-16:1 into *cis*-18:1 mentioned in a previous report [44]. In *E. coli*, both KAS I (*fabB*) and KAS II (*fabF*) have the ability for longer chain elongation of saturated fatty acids. KAS I is one of the key enzymes involved in the de novo synthesis of unsaturated fatty acids, while KAS II only extends the *cis*-hexadecenoyl to *cis*-octadecanoyl and does not participate in *de novo* synthesis of unsaturated acyl [45]. Combined with the studies above, we speculated that *fabB/F* was involved in the synthesis of unsaturated fatty acids. Leonard et al. [46] have found that the gene with KAS II activity in *Cuphea wrightii* could regulate FA chain length regulation, and was co-expressed with specific thioesterases, leading the content of C10:0 and C12:0 accumulation. Besides, Pidkowich et al. [47] proved that KAS II had the function of extention 16:0-ACP to 18:0-ACP in plant. Our results suggested that Synpccw7942_0537 (*fabB/F*) also performed the function of *fabF* and *fabB* in microalgae.

The antisense expression *fabH* also caused the change of fatty acid composition. As Table 3 shows, the content of C18 of *Synechococcus* sp. PCC 7942 containing pRL-Antisese-*fabH* decreased from 18.44% of the wild-type strain to 5.47% of antisense expression strains, with a decrease of 70.34%. At the same time, the content of C14 increased from 1.58% of the wild type to 3.63% of the antisense expression strain (Table 3), with an increase of 130% (*p* < 0.05). In addition, C16:2, C17:0, C17:1 and C18:2 content of *fabH* antisense expression strain all decreased, when C16:2 decreased from 0.27% to 0.07%, C18:2 decreased from 0.81% to 0.13%, C17 (17:0 and 17:1) fatty acids decreased from 1.3% to 0.63%. It is generally believed that *fabH* was related to the initiation of fatty acid elongation and a rate-limiting step of fatty acid synthesis [30]. González-mellado et al. [48] found that KAS III in sunflower had a new substrate specificity, which could participate in the synthesis of C6 to C10 fatty acids. Our present results showed that the C18 fatty acid decreased with the increase of C14 fatty acid when the gene *fabH* was inhibited, suggesting that the Synpccw7942_1455 (*fabH*) in *Synechococcus* sp. PCC 7942 may only participate in the carbon chain extension (C < 18) (not the first condensation step in the fatty acid synthesis circle) and had acyl-ACP carbon chain substrate specificity. While antisense expression *fabB/F* and *fabH* are able to redirect FA synthesis from long to short chain lengths, the increases in yield are still not as high as those achieved in plants and some bacteria. It is thought that high content of SCFAs could be poisonous for microalgae growth. Therefore, SCFAs accumulation is not only governed by KAS, but also by the tolerance of SCFAs concentration in microalgae cells.

### 2.4. Fatty Acid Compositions of fabB/F and fabH Deletion Mutants

The functions of Synpccw7942_0537 (*fabB/F*) and Synpccw7942_1455 (*fabH*) were further explored by gene deletion mutants. Table 4 showed that C16 was the main fatty acid component in the *fab(B/F)*^−^ mutant strain, accounting for about 60%–70% of the total fatty acid. The content of C8 increased by six times and C14 increased by 10.79%, while the content of C18 decreased by 9.87% (Table 4). In addition, the content of C15:0 fatty acid in the *fab(B/F)*^−^ mutant strain was 0.48% of the total fatty acids, which was not detected in the wild type (Table 4). These results suggested that the mutation of *fabB/F* gene might block fatty acid elongation. It was reported that the homologous protein of FabB could not found in some bacteria, but the homologous protein of FabF (FabY) had the dual functions of KAS I and KAS II, but this is not common [40]. The above results suggested that *fabB/F* in *Synechococcus* sp. PCC 7942 might have the function of fatty acid carbon chain elongation (C < 18), which were consistent with the functions of *fabB* and *fabF* in *E. coli*, where they carried out the elongation steps in fatty acid synthesis. The *fabB/F* in *Synechococcus* sp. PCC 7942 was also involved in the *de novo* synthesis of the unsaturated fatty acids of the microalgae and forming *cis*-11-octadecenoyl ACP by extending palmitoyl-ACP.

Table 4 also showed that the content of C18 decreased from 18.23% in the wild type to 16.91% in *fabH*^−^ mutant, with a decrease of 7.24%. Meanwhile, the content of C14 increased from 1.39% of the wild type to 1.48% of the mutant, with an increase of 6.47%, and 0.41% content of C15 was obtained (Table 4). These results indicated that the longer chain fatty acids decreased and shorter chain fatty acids increased when *fabH* was mutated. Previous studies showed that *fabH* was a rate-limiting step and also the first condensation step in the fatty acid synthesis circle in plants and bacteria [29,49]. The present results suggested that Synpccw7942_1455 (*fabH*) might only participate in the fatty acid carbon chain elongation (C < 18).

Nonetheless, the increment of SCFAs in mutants we observed was modest, far from our expectation. It was also reported that some enzyme had the activity of KAS III, with the domain similar to FabH, but whose molecular weight is larger than FabH, initializing fatty acid synthesis in *Pseudomonas aeruginosa* [46]. Therefore, it is assumed that *fab(B/F)*^−^ and *fabH*^−^ mutants were considered as strains deficient in *fabB/F* and *fabH* gene expression but not complete inactivation, or other proteins with similar activity executed corresponding functions (KAS III).

### 2.5. Complementation of E. coli

In order to further verify the functions of the related genes, we attempted to construct mutant strains of KAS I/II/III of *E. coli* and compared the fatty acid compositions of mutants with that of their complementation.

As shown in Table 5, the content of C12, C14 and C16 in *fab(B/F/H)*^−^ mutants of *E. coli* BL21 all increased as compared with that of the wild type. These results were consistent with the results that KAS III could catalyze acetyl-CoA with malonyl-CoA to generate 4:0-ACP, KAS I could catalyze 4:0-ACP to generate 16:0-ACP and KAS II could catalyze 16:0-ACP to generate 18:0-ACP as well as regulate the ratio of 16:0-ACP/18:0-ACP [24]. Mutation of the genes might lead to changes in fatty acids.

Synpccw7942_0537 (*fabB/F*) and Synpccw7942_1455 (*fabH*) from *Synechococcus* sp. PCC 7942 were used for *E. coli* mutants (*fabB/F/H*)^−^ complementation. The results show that fatty acid compositions of mutants were basically restored to that of the wild type after complementation (Table 5). These suggested that the Synpccw7942_0537 (*fabB/F*) gene in *Synechococcus* sp. PCC 7942 has similar functions of *fabB* and *fabF* of *E. coli* and was related to the elongation of fatty acids. Synpccw7942_1455 (*fabH*) has similar function of *fabH*, which is also related to the elongation of fatty acid in *Synechococcus* sp. PCC 7942.

### 2.6. Fluorescence Localization of the Related Proteins

In order to describe the SCFAs synthesis pathway in microalgae, cellular localization of Synpccw7942_0537 (*fabB/F*) and Synpccw7942_1455 (*fabH*) by co-expressing green fluorescence protein gene and potential genes were carried out. Figure 2 shows that the fusion expressed proteins of eGFP-FabB/F and eGFP-FabH in *Synechococcus* sp. PCC 7942 were evenly dispersed on the cell membrane, indicating that the enzymes corresponding to *fabB/F* and *fabH* were located on the cell membrane. It is the first time to show the cell location of KASs.

Based on the above studies, a possible pathway for the SCFAs synthesis in *Synechococcus* sp. PCC 7942 is proposed. As shown in Figure 3, CO_2_ enters the microalgal cell and finally forms malonyl-ACP through de novo fatty acids synthesis, then goes into the fatty acid synthesis pathway, where KAS I/II (*fabB/F*) and KAS III (*fabH*) located on the cell membrane could catalyze condensation reactions to accumulate different chain length fatty acids less than 18 carbon atoms. Previous research found that fatty acids synthetic system started from acetyl-CoA carboxylase (ACCase), catalyzing the conversion of acyl-CoA to malonyl CoA, which was thought to be a limiting step [50]. KAS III had also been proved to be a limiting step in microalgae fatty acid biosynthesis. The above two limiting steps may affect the fixation of CO_2_ in microalgae fatty acid biosynthesis. In the whole photosynthetic CO_2_ fixation process, to increase the lipid productivity, ACCase and KAS III are the best options available for operation. Overexpression of *ACCase* and *KAS III* genes has succeeded to regulate the lipid synthesis [29,51,52].

## 3. Materials and Methods

### 3.1. Microalgae Cultures

*Synechococcus* sp. PCC 7942 used in the study was provided by Prof. Dingji Shi (Shanghai Ocean University, Shanghai, China). Cells were preserved in the BG11 medium. The initial OD_730_ was 0.1, the initial pH was 7.8–8.0. The cultivation of *Synechococcus* sp. PCC 7942 was bubbled under 27 ± 1 °C and 140 mmol m^−2^ s^−1^ in 1 L Erlenmeyer flask with 500 mL working volume of BG11 medium. BG11 medium consists of (1 L) 1.5 g NaNO_3_; 0.03 g K_2_HPO_4_; 0.075 g MgSO_4_·2H_2_O; 0.036 g CaCl_2_·2H_2_O; 0.006 g citric acid; 0.006 g ferric ammonium citrate; 0.001 g EDTA; 0.02 g Na_2_CO_3_ and 1 mL micronutrient solution. The microelement solution consists of (1 L) 2.86 g H_3_BO_3_; 1.81 g MnCl_2_·4H_2_O; 0.222 g ZnSO_4_·7H_2_O; 0.39 g NaMoO_4_·5H_2_O; 0.079 g CuSO_4_·5H_2_O and 0.0494 g Co(NO_3_)_2_·6H_2_O.

Cell growth in BG11 medium was set as the normal growth condition. Different cerulenin concentrations were set as follows: 0 g/L, 1 g/L, 2.5 g/L, 5 g/L and 7.5 g/L. Stock solution of cerulenin (100 mg/L, preserved in ethyl alcohol, −20 °C) was added to BG11 medium with the final concentrations at 0 day.

### 3.2. Genomic DNA Extraction and PCR Analysis

Genomic DNA of *Synechococcus* sp. PCC 7942 was isolated as described previously by Kuo et al. [30]. Genomic DNA of *Synechococcus* sp. PCC 7942 transformant was prepared from exponential growth phase cultures using a procedure based on cetyltrimethylammonium bromide (CTAB) method. PCR analysis of transformants was carried out with genomic DNA as a template using primers as shown in Appendix A. PCR amplification was performed for 30 cycles of 95 °C for 1 min, 56 °C for 30 s, and 72 °C for 30 s, followed by 72 °C for 10 min.

### 3.3. Sense and Antisense Expression Vector Construction

Bioinformatics showed that two enzymes in *Synechococcus* sp. PCC 7942 were annotated with β-ketoacyl ACP synthase activity. Alignments were proved that the sequence of Synpccw7942_0537 (Gene ID: 3774775) is similar to *fabB* and *fabF* in plants and bacteria, while the sequence of Synpccw7942_1455 (Gene ID: 3773627) is similar to *fabH* in plants and bacteria. In this experiment, we labeled them as *fabF/B* (Synpccw7942_0537) and *fabH* (Synpccw7942_1455), respectively.

The sequence Synpcc7942_0537 (*fabB/F*) and Synpcc7942_1455 (*fabH*) of *Synechococcus* sp. PCC 7942 were amplified by PCR from total DNA, using 7942-anti-fabB/F-F, 7942-anti-fabB/F-R and 7942-anti-fabH-F, 7942-anti-fabH-R (7942-sen-fabB/F-F, 7942-sen-fabB/F-R and 7942-sen-fabH-F, 7942-sen-fabH-R) primers (Appendix A) and then sub-cloned into pMD19-T vector (pMD19-T Vector Cloning Kit, TaKaRa Biotech Co., Dalian, China), resulting in pMD19T-anti*fabB/F* (pMD19T-sen*fabB/F*) and pMD19T-anti*fabH* (pMD19T-sen*fabH*) vectors. The amplified DNA fragments were digested with restriction enzymes XhoI and KpnI, and ligated into the corresponding sites of pRL489 vector to generate the antisense expression vectors (a) pRL-sen*fabB/F*, pRL-sen*fabH* and antisense expression vectors (b) pRL-antisen*fabB/F*, pRL-antisen*fabH*, using kanamycin as a selectable marker. These constructs were used for the natural transformation, then selected transformants by gradient antibiotics screening. Plasmids were propagated and purified using a standard procedure.

### 3.4. Deletion Mutant Construction

The plasmid pUC118 was constructed for gene deletion mutants of Synpcc7942_0537 (*fabB/F*) and Synpcc7942_1455 (*fabH*). DNA fragments upstream and downstream of Synpcc7942_0537 (*fabB/F*) and Synpcc7942_1455 (*fabH*) were amplified from *Synechococcus* sp. PCC 7942 genomic DNA using the primers in Appendix A. Chloromycetin gene was amplified from pET28a. These upstream and downstream DNA fragments and chloromycetin gene were amplified by using overlap PCR to form *fabB/F*-cat and *fabH*-cat fragments, and inserted into pUC118, replacing the corresponding regions. The resulting plasmids (pUC118) contains homologous regions for gene deletion mutants of Synpcc7942_0537 (*fabB/F*) and Synpcc7942_1455 (*fabH*), and chloromycetin antibiotic resistance was used for selection. The deletion mutants were further confirmed by DNA sequence analysis (Appendix A).

### 3.5. Fusion Expression Vector of Green Fluorescence Protein Construction

The plasmid pRL489 was used for fusion expression vectors of green fluorescence protein gene and Synpcc7942_0537 (*fabB/F*) and Synpcc7942_1455 (*fabH*). DNA sequences of Synpcc7942_0537 (*fabB/F*) and Synpcc7942_1455 (*fabH*) were amplified from *Synechococcus* sp. PCC 7942 genomic DNA using the primers in Appendix A. Enhanced green fluorescence protein gene (*egfp*) fragment was from plasmid preserved in our laboratory. Synpcc7942_0537 (*fabB/F*) and Synpcc7942_1455 (*fabH*) and *egfp* gene were amplified by using overlap PCR to form egfp-fabB/F and egfp-fabH, and inserted into the pRL489, resulting pRL-egfp-*fabB/F* and pRL-egfp-*fabH*. Kanamycin antibiotic resistance was used for selection.

### 3.6. Liquid Culture and Screening of Transgenic Microalgae

Monoalgal colonies screened under antibiotic were cultured in BG11 medium containing kanamycin sulfate. Initially, transgenic microalgae was cultured in BG11 medium containing 25 g/L of kanamycin, and then the concentration of kanamycin (50 g/L, 100 g/L, 150 g/L, 200 g/L, 250 g/L, and 300 g/L) was gradually increased. When the concentration of antibiotic was increased to 300 g/L, wild type strains completely died under the pressure of antibiotic.

### 3.7. RNA Extraction and cDNA Synthesis

Total RNA extraction was treated with Trizol reagent (Sangon Biotech, Shanghai, China). Microalgal cells were harvested at different times (2 day intervals) by centrifugation. Approximately, microalgal cells (0.1 g) were ground in liquid nitrogen to powder before adding 1 mL Trizol reagent. After centrifugation at 12,000 r.p.m. for 10 min at 48 °C, 200 mL chloroform was added and mixed thoroughly. The sample was centrifuged at 12,000 r.p.m. for 15 min, then 450 mL of the uppermost layer was transferred into a fresh tube. Isopropanol (450 mL) was added to precipitated RNA and then centrifuged at 12,000 r.p.m. for 15 min, washed with 1 mL ethanol (75%, *v*/*v*) twice, dissolved in 30 mL diethyl pyrocarbonate treated distilled water. M-MLV (Moloney Murine Leukemia Virus) reverse transcriptase (TaKaRa Biotech Co., Dalian, China) was used to synthesize the cDNA. Two fatty acid synthetase genes (*fabF/B* and *fabH*) were investigated in this study. Genome annotation identified putative components of fatty acid biosynthetic pathway including *fabF/B* and *fabH*. Primers were designed according to the highly conserved regions obtained from the alignment of public database and the deduced amino acid sequences. The primers used in the experiment were shown in Appendix A. After amplification, the PCR products were purified and cloned into pMD-19T vector for sequencing.

### 3.8. Quantitative Real-Time PCR

Real-time PCR was carried out on CFX96 Touch Real-Time PCR Detection System (Bio-Rad, Hercules, CA, USA) using SYBR Premix Ex TaqTMII (TaKaRa Biotech Co., Dalian, China). The reference gene was used as an internal standard, amplified with the *rnpB*-F/*rnpB*-R primers (Appendix A). Real-time PCR was conducted following the procedure: 95 °C for 30 s before performing 40 cycles of 95 °C for 5 s and 60 °C for 45 s, and a melting step at 60 °C–95 °C. PCR efficiency of each gene was calculated by relative standard curve using sequential dilutions of the cDNA. The 2^−ΔΔCT^ method [53] was applied for the target gene expression calculation.

### 3.9. Lipid Extraction and Fatty Acid Analysis

Microalgal cells were harvested by centrifugation after cultivation. The total lipids were extracted according to the following method. Lyophilized microalgae powder (0.2 g) was pulverized in a mortar and extracted using 5 mL solvent mixture of chloroform:methanol (2:1, *v*/*v*). After shaking for 10 min, the samples were centrifuged (5804R, Eppendorf, Germany) at 10,000 r.p.m. for 10 min. The procedure was repeated three times to make sure the lipids were extracted completely. The solvent phase was transferred by pipette and evaporated in a water bath at 60 °C.

The fatty acids component of the lipids was analyzed by GC-linked mass spectrometry (GC-MS). Fatty acid methyl esters were obtained by acidic transesterification of the lipids. After reaction at 90 °C for 2 h, 2.5 mL hexane and 1 mL water were added to the sample, then vibrated gently and centrifuged. One milliliter of the organic upper phase was injected into an Auto System XL GC/Turbo Mass MS (Perkin Elmer, Germany) using a DB-5MS (5% phenyl)-methylpolysiloxane nonpolar column (30 m × 0.25 mm × 0.25 mm). At the beginning, the column temperature was kept at 60 °C for 4 min, then increased to 220 °C, and finally reached to 280 °C with a temperature gradient of 10 °C min, maintained for 10 min.

## 4. Conclusions

This study demonstrated that SCFAs synthesis of *Synechococcus* sp. PCC 7942 relied on β-ketoacyl ACP synthase (KAS), where Synpccw7942_0537 (*fabB/F*) had the function of short chain fatty acids elongation (C < 18) and elongation of C16:1 to C18:1, and Synpccw7942_1455 (*fabH*) could participate the SCFAs elongation (C < 18). Moreover, cerulenin, which could inhibit the expression level of Synpccw7942_0537 (*fabB/F*) and Synpccw7942_1455 (*fabH*), led to SFCAs accumulation. Based on these results, the SCFAs biosynthesis pathway in microalgae was proposed.

## Figures and Tables

**Figure 1 marinedrugs-17-00255-f001:**
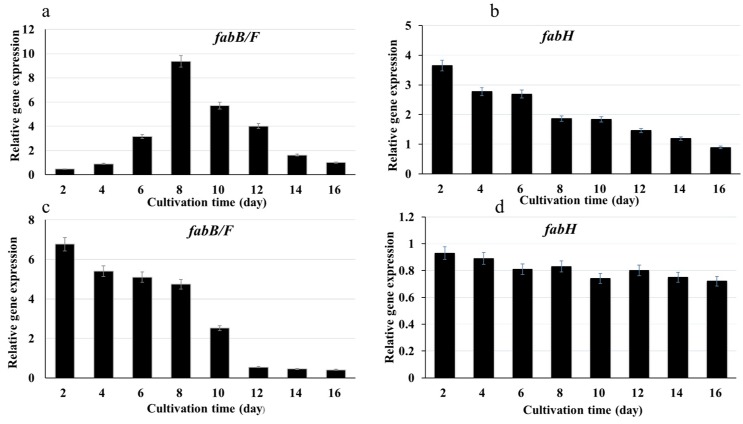
Dynamic change of expression levels of two related genes in *Synechococcus* sp. PCC 7942 under different conditions. (**a**) Expressions of Synpccw7942_0537 (*fabB/F*) under different culture time periods; (**b**) Expressions of Synpccw7942_1455 (*fabH*) under different culture time periods; (**c**) Expressions of Synpccw7942_0537 (*fabB/F*) under cerulenin concentration of 7.5 g/L; (**d**) Expressions of Synpccw7942_1455 (*fabH*) under cerulenin concentration of 7.5 g/L. The expression levels of genes Synpccw7942_0537 (*fabB/F*) and Synpccw7942_1455 (*fabH*) at 16 day (**a**,**b**) and cerulenin concentration of 0 g/L (**c**,**d**) were set to 1.

**Figure 2 marinedrugs-17-00255-f002:**
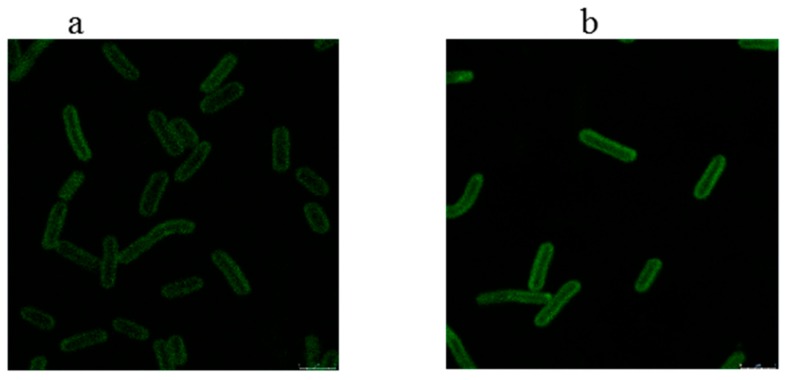
The fusion expressed proteins of FabB/F-eGFP (**a**) and FabH-eGFP (**b**) in *Synechococcus* sp. PCC 7942 observed under the confocal laser fluorescence microscope. (**a**) FabB/F labeled with eGFP in *Synechococcus* sp. PCC 7942 for fluorescence localization; (**b**) FabH labeled with eGFP in *Synechococcus* sp. PCC 7942 for fluorescence localization.

**Figure 3 marinedrugs-17-00255-f003:**
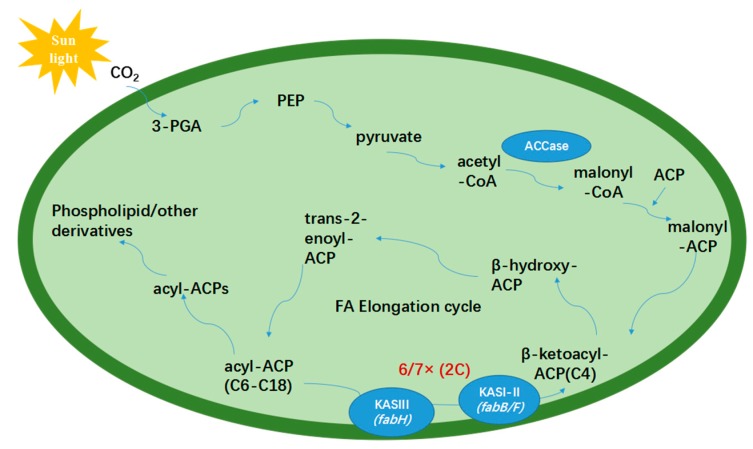
The proposed synthesis pathway of short carbon fatty acids (SCFAs) in *Synechococcus* sp. PCC 7942.

**Table 1 marinedrugs-17-00255-t001:** Fatty acid profile of *Synechococcus* sp. PCC 7942 under different culture time.

Fatty Acid (%)	Time (day)									
	2	4	6	8	10	12	14	16	18	20
12:0	0.02 ± 0.01	ND	0.02 ± 0.00	ND	ND	ND	ND	ND	ND	ND
14:0	0.63 ± 0.03	0.74 ± 0.15	0.56 ± 0.06	0.45 ± 0.03	0.62 ± 0.03	0.27 ± 0.03	0.33 ± 0.01	0.42 ± 0.03	0.34 ± 0.03	0.36 ± 0.03
14:1	0.98 ± 0.02	1.70 ± 0.19	1.17 ± 0.19	0.87 ± 0.10	0.90 ± 0.06	0.48 ± 0.03	0.55 ± 0.04	0.67 ± 0.06	0.57 ± 0.03	0.60 ± 0.06
16:0	42.14 ± 3.96	39.98 ± 2.67	45.42 ± 2.64	41.71 ± 2.22	38.65 ± 2.68	47.93 ± 2.00	45.41 ± 2.11	46.56 ± 2.04	46.63 ± 3.08	46.20 ± 2.05
16:1	34.72 ± 2.04	38.20 ± 1.89	36.80 ± 1.74	31.03 ± 1.25	31.32 ± 0.71	26.96 ± 1.71	31.94 ± 1.87	29.89 ± 0.89	29.60 ± 1.60	30.48 ± 1.89
16:2	ND	ND	ND	ND	ND	ND	ND	ND	0.13 ± 0.03	0.13 ± 0.02
17:0	0.13 ± 0.03	0.22 ± 0.03	0.09 ± 0.03	0.12 ± 0.03	0.11 ± 0.02	0.18 ± 0.04	0.11 ± 0.03	0.10 ± 0.00	0.13 ± 0.01	0.11 ± 0.00
17:1	0.22 ± 0.01	0.41 ± 0.01	0.14 ± 0.03	0.24 ± 0.05	0.27 ± 0.02	0.30 ± 0.00	0.33 ± 0.00	0.34 ± 0.03	0.32 ± 0.01	0.33 ± 0.03
18:0	1.70 ± 0.34	3.00 ± 0.21	0.24 ± 0.05	2.49 ± 0.24	3.11 ± 0.25	2.57 ± 0.25	1.89 ± 0.37	1.08 ± 0.11	1.85 ± 0.11	1.61 ± 0.03
18:1	2.51 ± 0.33	3.89 ± 0.29	1.80 ± 0.08	3.79 ± 0.11	4.48 ± 0.31	7.14 ± 0.75	6.42 ± 0.57	4.90 ± 0.36	6.15 ± 0.36	5.82 ± 0.23
18:2	0.31 ± 0.047	0.24 ± 0.09	3.16 ± 0.32	0.34 ± 0.04	0.39 ± 0.03	0.35 ± 0.02	0.68 ± 0.03	0.21 ± 0.03	0.41 ± 0.02	0.43 ± 0.03
20:0	0.04 ± 0.00	0.03 ± 0.00	0.04 ± 0.00	0.08 ± 0.03	0.12 ± 0.01	0.05 ± 0.01	ND	ND	0.06 ± 0.01	0.13 ± 0.02
22:0	0.04 ± 0.01	0.04 ± 0.01	0.05 ± 0.01	ND	0.19 ± 0.01	ND	ND	ND	0.26 ± 0.02	0.27 ± 0.04
22:1	0.05 ± 0.00	0.04 ± 0.00	0.04 ± 0.00	ND	0.11 ± 0.01	0.10 ± 0.00	0.07 ± 0.00	ND	ND	0.06 ± 0.00

**Table 2 marinedrugs-17-00255-t002:** Effects of different cerulenin concentrations on fatty acid profile in *Synechococcus* sp. PCC 7942 after 16 days cultivation.

Fatty Acid (%)	Cerulenin Concentration (g/L)				
	0	1	2.5	5	7.5
12:0	0.04 ± 0.00	0.05 ± 0.00	0.04 ± 0.01	0.07 ± 0.00	0.09 ± 0.01
14:0	0.71 ± 0.08	0.98 ± 0.09	0.77 ± 0.04	0.84 ± 0.09	1.25 ± 0.31
14:1	0.87 ± 0.15	1.1 ± 0.26	0.82 ± 0.08	1.35 ± 0.17	1.5 ± 0.31
15:0	0.07 ± 0.02	0.07 ± 0.01	ND	ND	ND
16:0	31.47 ± 2.48	41.74 ± 1.85	34.76 ± 1.68	35.43 ± 1.53	31.84 ± 2.46
16:1	31.75 ± 2.39	32.97 ± 2.56	25.66 ± 2.90	26.57 ± 3.38	25.43 ± 2.45
16:2	0.27 ± 0.03	0.07 ± 0.02	ND	0.06 ± 0.01	0.08 ± 0.01
17:0	0.49 ± 0.03	0.16 ± 0.04	0.16 ± 0.02	0.21 ± 0.02	0.47 ± 0.01
17:1	0.81 ± 0.12	0.43 ± 0.02	0.52 ± 0.06	0.51 ± 0.04	0.76 ± 0.11
18:0	1.79 ± 0.10	2.14 ± 0.42	3.33 ± 0.33	2.7 ± 0.15	3.57 ± 0.26
18:1	15.84 ± 0.98	8.02 ± 0.14	14.05 ± 1.47	12.13 ± 1.49	5.69 ± 0.04
18:2	0.81 ± 0.08	1.15 ± 0.11	0.69 ± 0.03	0.9 ± 0.10	4.44 ± 0.27
20:0	0.04 ± 0.03	0.05 ± 0.00	ND	ND	ND

**Table 3 marinedrugs-17-00255-t003:** Fatty acid profile of sense and antisense expression of Synpccw7942_0537 (*fabB/F*) and Synpccw7942_1455 (*fabH*) in *Synechococcus* sp. PCC 7942 after 16 days cultivation.

Fatty Acid (%)	Wild Type ^a^	Sense-*fabB/F* ^b^	Sense-*fabH* ^c^	Antisense-*fabB/F* ^d^	Antisense-*fabH* ^e^
C12:0	0.04 ± 0.01	0.06 ± 0.01	0.05 ± 0.00	0.06 ± 0.00	0.05 ± 0.01
C14:0	0.71 ± 0.08	0.71 ± 0.06	0.65 ± 0.04	1.09 ± 0.16	1.35 ± 0.35
C14:1	0.87 ± 0.07	0.75 ± 0.07	0.66 ± 0.03	1.52 ± 0.12	2.28 ± 0.46
C15:0	0.07 ± 0.00	0.29 ± 0.03	0.44 ± 0.02	0.10 ± 0.00	0.06 ± 0.01
C16:0	31.47 ± 0.63	32.04 ± 1.38	38.06 ± 0.55	34.99 ± 0.72	36.09 ± 2.31
C16:1	31.75 ± 2.40	33.16 ± 0.53	34.26 ± 1.68	37.2 ± 1.35	39.4 ± 1.19
C16:2	0.27 ± 0.02	0.16 ± 0.03	0.05 ± 0.01	0.13 ± 0.02	0.07 ± 0.02
C17:0	0.49 ± 0.02	0.32 ± 0.05	0.39 ± 0.06	0.10 ± 0.01	0.24 ± 0.03
C17:1	0.81 ± 0.01	0.56 ± 0.02	0.49 ± 0.02	0.77 ± 0.01	0.39 ± 0.03
C18:0	1.79 ± 0.21	6.83 ± 0.74	8.3 ± 0.36	0.72 ± 0.11	0.82 ± 0.03
C18:1	15.84 ± 2.00	8.05 ± 0.10	8.49 ± 0.32	1.25 ± 0.05	4.52 ± 0.65
C18:2	0.81 ± 0.08	3.33 ± 0.20	0.33 ± 0.03	0.18 ± 0.07	0.13 ± 0.03
C20:0	0.04 ± 0.01	0.06 ± 0.00	ND	0.72 ± 0.03	ND

^a^ Wild type strain of *Synechococcus* sp. PCC 7942; ^b^
*Synechococcus* sp. 7942 harboring pRL-Sense-*fabB/F* (Synpccw7942_0537); ^c^
*Synechococcus* sp. 7942 harboring pRL-Sense-*fabH* (Synpccw7942_1455); ^d^
*Synechococcus* sp. 7942 harboring pRL-Antisense-*fabB/F* (Synpccw7942_0537); ^e^
*Synechococcus* sp. 7942 harboring pRL-Antisense-*fabH* (Synpccw7942_1455).

**Table 4 marinedrugs-17-00255-t004:** Fatty acid profile of Synpccw7942_0537 (*fabB/F*) and Synpccw7942_1455 (*fabH*) deletion mutants of *Synechococcus* sp. PCC 7942 after 16 days cultivation.

Fatty Acid (%)	Wild Type ^a^	*fab(B/F)^−^*	*fabH^−^*
C8:0	0.01 ± 0.00	0.06 ± 0.02	0.02 ± 0.00
C14:0	0.54 ± 0.04	0.63 ± 0.02	0.67 ± 0.07
C14:1	0.85 ± 0.02	0.91 ± 0.02	0.81 ± 0.00
C15:0	ND	0.48 ± 0.05	0.41 ± 0.03
C16:0	34.30 ± 0.25	36.21 ± 0.40	33.16 ± 0.04
C16:1	32.43 ± 0.63	30.93 ± 0.32	34.86 ± 0.70
C16:2	0.51 ± 0.01	0.64 ± 0.04	0.06 ± 0.00
C17:0	0.26 ± 0.06	0.69 ± 0.03	0.38 ± 0.05
C17:1	0.78 ± 0.14	0.74 ± 0.04	0.52 ± 0.05
C18:0	1.85 ± 0.03	1.93 ± 0.03	6.23 ± 0.58
C18:1	15.69 ± 0.90	13.71 ± 0.39	8.17 ± 0.35
C18:2	0.69 ± 0.05	0.79 ± 0.03	2.51 ± 0.54
C20:0	0.05 ± 0.00	0.02 ± 0.01	0.04 ± 0.00

^a^ Wild type strain of *Synechococcus* sp. PCC 7942.

**Table 5 marinedrugs-17-00255-t005:** Fatty acid profile of wild type, mutation and complementary mutation of *E. coli*.

Fatty Acid (%)	*E. coli* BL21	*fabB^−^*	*fabF^−^*	*fabH^−^*	Com-*fabB* ^a^	Com-*fabF* ^b^	Com-*fabH* ^c^
C12:0	0.23 ± 0.02	0.3 ± 0.00	0.72 ± 0.10	0.78 ± 0.13	0.31 ± 0.12	0.50 ± 0.10	0.23 ± 0.01
C13:0	1.33 ± 0.11	ND	ND	0.10 ± 0.00	1.19 ± 0.00	0.94 ± 0.23	1.87 ± 0.34
C14:0	2.88 ± 0.28	3.04 ± 0.23	3.69 ± 0.29	3.01 ± 0.46	2.88 ± 0.46	3.03 ± 0.33	3.87 ± 0.58
C14:1	ND	3.23 ± 0.20	0.21 ± 0.01	0.30 ± 0.00	ND	ND	ND
C15:0	8.25 ± 0.22	0.57 ± 0.03	1.02 ± 0.07	0.72 ± 0.15	6.32 ± 0.15	5.97 ± 0.55	12.45 ± 2.02
C16:0	33.98 ± 3.83	50.51 ± 2.75	48.59 ± 3.11	50.23 ± 3.64	33.82 ± 0.64	33.37 ± 2.50	34.76 ± 1.55
C16:1	2.78 ± 0.30	9.21 ± 0.22	9.94 ± 1.60	8.21 ± 0.94	2.60 ± 0.94	2.66 ± 0.35	3.08 ± 0.19
C17:0	5.02 ± 0.21	0.44 ± 0.04	0.60 ± 0.18	0.51 ± 0.07	3.56 ± 0.07	3.94 ± 0.42	7.57 ± 0.67
C17:1	2.69 ± 0.17	2.97 ± 0.31	2.46 ± 0.38	5.18 ± 0.56	3.71 ± 0.56	2.36 ± 0.34	1.99 ± 0.12
C18:0	2.30 ± 0.16	1.40 ± 0.15	2.93 ± 0.54	2.37 ± 0.40	2.62 ± 0.40	2.07 ± 0.23	2.20 ± 0.24
C18:1	20.57 ± 0.94	6.61 ± 0.51	7.12 ± 0.29	5.39 ± 0.68	20.51 ± 0.68	20.62 ± 2.83	5.55 ± 0.77
C18:2	0.17 ± 0.02	0.34 ± 0.02	0.52 ± 0.02	0.37 ± 0.03	0.16 ± 0.03	0.17 ± 0.03	8.36 ± 0.51
C19:0	1.11 ± 0.11	1.34 ± 0.05	1.22 ± 0.12	2.83 ± 0.29	5.20 ± 0.29	1.25 ± 0.15	0.96 ± 0.14
C20:0	0.03 ± 0.01	0.02 ± 0.00	0.04 ± 0.01	0.03 ± 0.00	0.04 ± 0.00	0.03 ± 0.00	0.03 ± 0.01
C20:1	1.59 ± 0.17	0.41 ± 0.04	0.73 ± 0.03	0.31 ± 0.03	1.76 ± 0.03	1.98 ± 0.17	1.03 ± 0.03
C21:0	0.01 ± 0.00	ND	0.02 ± 0.00	0.02 ± 0.01	0.01 ± 0.01	ND	ND
C22:0	0.12 ± 0.01	0.03 ± 0.00	0.14 ± 0.02	0.07 ± 0.01	0.11 ± 0.01	0.09 ± 0.02	0.16 ± 0.01
C22:1	0.05 ± 0.00	0.02 ± 0.01	ND	ND	0.08 ± 0.03	0.02 ± 0.00	ND
C23:0	0.10 ± 0.00	0.03 ± 0.01	0.05 ± 0.00	0.04 ± 0.01	0.06 ± 0.01	0.07 ± 0.00	0.17 ± 0.00
C24:0	0.09 ± 0.00	0.02 ± 0.00	0.05 ± 0.00	0.04 ± 0.01	0.06 ± 0.01	0.05 ± 0.00	0.15 ± 0.02

^a^*E. coli* BL21 *fabB* mutant complement with *fabB/F* of *Synechococcus* sp. PCC 7942. ^b^
*E. coli* BL21 *fabF* mutant complement with *fabB/F* of *Synechococcus* sp. PCC 7942. ^c^
*E. coli* BL21 *fabH* mutant complement with *fabH* of *Synechococcus* sp. PCC 7942.

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
