# Peer review of "Short Chain Fatty Acid Biosynthesis in Microalgae Synechococcus sp. PCC 7942"

_marinedrugs, 2019, doi:10.3390/md17050255_

Round 1

Reviewer 2 Report

the manuscript was modified according to my suggestions. It can be accepted for publication in Marine Drugs journal.

Reviewer 3 Report

The article revisions have significantly improved the quality of paper.

This manuscript is a resubmission of an earlier submission MANUSCRIPT # 473901. The following is a list of the peer review reports and author responses from that submission.

Round 1

Reviewer 1 Report 

REVIEW: This manuscript reports several approaches to assessing the utility of the marine cyanobacteria Synechococcus PCC7942 as an alternate source for mass production of short chain fatty acids (SCFAs, C6-C14 chain lengths) for nutraceutical related applications.  The authors follow a logical plan in this assessment: 1) determine the dynamics of SCFA accumulation during a culture cycle of this cyanobacterium, 2) determine potential culture manipulations to increase yields of SCFA through a) use of FA synthesis inhibitor and antifungal agent cerulenin, and b) identification of enzymatic targets regulating SCFA pool sizes, which the authors hypothesize are primarily control via the activities of FabB/F and FabH gene products.  While the objectives of the paper are clear and represent a step forward in identifying reliable SCFA sources for industrial applications the presentation and lack of statistical information determining the significance of treatment effects observed prevents this reviewer from recommending this manuscript for publication in its present form. 

Both English phrasing and inconsistencies in results interpretation need to be improved starting with the title: “Short chain fatty acidb biosynthesis relies on β- ketoacyl ACP synthases in …….”

This reviewer feels that presentation would be more logical to flip flop the discussion and first two tables with FA composition during culture cycle as Table 1 and the effects of cerulenin as table 2.  Accompanying this change should also be some discussion of the significance of small subpercentage changes in relative pools sizes of the FA, ie how variable is the composition especially for the smaller targeted SCFA pools.

Also while the use of cerulenin as a means to manipulate KASI-III (fabB, F, H) activity is great, the fact that high concentrations of this antimycotic have to be employed to ‘significantly’ enhance SCFA pools should be addressed as this is an expensive reagent (>$20/mg) and potentially adding tremendous cost to the SCFA endproduct production. 

The reviewer found it difficult to interpret this %FA pool sizes without information on the cellular biomass changes during the growth cycle at minimum there should be some indication of yield of FA per dry weight, chlorophyll, or cell# throughout culture cycle.  This is critical in the discussion of Figure 1 where the sift in expression patterns may simply reflect growth inhibition by cerulenin. 

Many of the changes in FA pool representation discussed with these tables and as a result of antisense or deletion manipulation of fabB/F and fabH homologs are being biased by one member of the authors FA chain length summaries ie 16:1 and !8:1 with little change in other forms. 

In terms of effects of cerulenin on relative transcript pool sizes (can’t say “expression” as no growth rate data provided).   The authors on line 164 state: “indicating that the influence of cerulenin on the expression of fabH was weak” is totally off the mark.  fabH abundance in Fig1d is less than ¼ the level in control cultures, being more equivalent to d16 levels in untreated cultures.  Assuming that experimental cultures were started from older stock cultures (e.g. day 16), fabH expression only decreases or is diluted from this level during cerulenin (i.e. fabH expression is inhibited by this treatment).

In terms of antisense and deletion mutant experiments in tables 3 and 4 the reader has no clue as to what growth stage (log, stationary, senescent cultures) are being compared.  Also their own indication on line 202 that “antisense fragment could inhibit the expression level of fabB/F” should be clearly demonstrated by qPCR comparisons of transcript abundance for all of your transformed lines with wild type strain.  This should easily be supplied as supplemental information. 

The review and reader would benefit from some discussion of why the fabB/F and particularly fabH knockouts in Synechoccocus are not lethal? Were are they getting the FA to continue growth from, certainly not BG11.  If fabH regulates entry into FA elongation cycle then lack of fabH should dramatical knock down at least total FA accumulation relative to cyanobacterial biomass…..  Again it should be simple to provide data on integration of your knockout cassette into the genome….. if its not in the correct place (which would be consistent with the data presented), then its not a knockout mutation…

Ln 254 and beyond, use complementation rather than “heter-complementation”

Authors’ responses:

REVIEW: This manuscript reports several approaches to assessing the utility of the marine cyanobacteria Synechococcus PCC7942 as an alternate source for mass production of short chain fatty acids (SCFAs, C6-C14 chain lengths) for nutraceutical related applications.  The authors follow a logical plan in this assessment: 1) determine the dynamics of SCFA accumulation during a culture cycle of this cyanobacterium, 2) determine potential culture manipulations to increase yields of SCFA through a) use of FA synthesis inhibitor and antifungal agent cerulenin, and b) identification of enzymatic targets regulating SCFA pool sizes, which the authors hypothesize are primarily control via the activities of FabB/F and FabH gene products.  While the objectives of the paper are clear and represent a step forward in identifying reliable SCFA sources for industrial applications the presentation and lack of statistical information determining the significance of treatment effects observed prevents this reviewer from recommending this manuscript for publication in its present form.

Thanks for the comments! According to the reviewer’s suggestion, we have revised the manuscript carefully. We have improved the presentation and added statistical information to the results (Figures, Tables, paragraph 4 of section 2.1, paragraph 3 and 4 of section 2.3).

Both English phrasing and inconsistencies in results interpretation need to be improved starting with the title: “Short chain fatty acids biosynthesis relies on β- ketoacyl ACP synthases in …….”

Thanks for the comments! We have improved English phrasing and inconsistencies in results interpretation in the manuscript. The title was revised to “Short chain fatty acid biosynthesis in microalgae Synechococcus sp. PCC 7942”. We highlight in yellow the changes made in revised version.

This reviewer feels that presentation would be more logical to flip flop the discussion and first two tables with FA composition during culture cycle as Table 1 and the effects of cerulenin as table 2.  Accompanying this change should also be some discussion of the significance of small subpercentage changes in relative pools sizes of the FA, ie how variable is the composition especially for the smaller targeted SCFA pools.

Thanks for your kind suggestion! As suggested, we have flipped flop the discussion and first two tables with FA composition during culture cycle as Table 1 and the effects of cerulenin as Table 2. Some discussion of small subpercentage changes in relative pools sizes of the FA, especially the composition for the smaller targeted SCFA pools has been added. These revisions are in the paragraph 2-4 of section 2.1.

Also while the use of cerulenin as a means to manipulate KASI-III (fabB, F, H) activity is great, the fact that high concentrations of this antimycotic have to be employed to ‘significantly’ enhance SCFA pools should be addressed as this is an expensive reagent (>$20/mg) and potentially adding tremendous cost to the SCFA endproduct production. 

Thanks for the comments! We agreed with the reviewer that high concentrations of cerulenin could significantly enhance SCFA pools, while the high price of cerulenin potentially added tremendous cost to the SCFA end product production. It is reported that cerulenin have specific inhibition on KAS I and KAS II, which are responsible for the carbon chain elongation of C4-C16 fatty acids in bacteria. Therefore, we employed cerulenin to investigate the functions of the related genes in the study. There may be some misunderstandings caused by our poor English expression. We have revised and improved the English use in the manuscript. These revisions are in Abstract and the paragraph 3 of section 2.1.

The reviewer found it difficult to interpret this %FA pool sizes without information on the cellular biomass changes during the growth cycle at minimum there should be some indication of yield of FA per dry weight, chlorophyll, or cell# throughout culture cycle.  This is critical in the discussion of Figure 1 where the sift in expression patterns may simply reflect growth inhibition by cerulenin. 

Thanks for the comments! This is a valuable suggestion. In fact, cerulenin could inhibit the growth of Synechococcus sp. PCC7942 (following Figure S1).When the cerulenin concentration was between 0 g/L and 7.5 g/L, the OD730 value of the same culture time decreased with the increase of the concentration of cerulenin, which indicated that the growth of microalgae was affected by the concentration of cerulenin. When the cerulenin concentration was gradually increased, the growth of microalgae was inhibited or even died. At 7.5 g/L cerulenin concentration, the OD730 value was the lowest, which was about one third of that at 0 g/L. We provided the growth information as supplementary Figure S1.

The %FA pool sizes in the study were calculated based on the total fatty acids compositions detected by GC-MS. Fatty acid methyl esters were obtained by acidic transesterification of the lipids. For different samples, the same weight of lipids (0.1 g) was used in transesterification reactions. After transesterification, the same volume (l mL) of the sample was used for GC-MS analysis. Thus, the %FA pool sizes based on the total fatty acids compositions could simply reflect the changes of fatty acids compositions under different culture conditions. Some of the above information has been added to the last paragraph of section 2.2 and the paragraph 2 of section 3.9.

The lipids content and the fatty acid composition and content of microalgae can vary significantly under different culture conditions. Growth inhibition by cerulenin could definitely affect the fatty acid composition and content of Synechococcussp. PCC 7942. The accurate content of fatty acids can also be obtained by adding C19 as internal standard in transesterification reaction. The reviewer’s comments will be helpful for our subsequent experiments.

The expression patterns of two related genes in Synechococcus sp. PCC 7942 may be the result of both growth and gene inhibition by cerulenin. This was discussed at the following point 6.

Please see Figure S1.

Many of the changes in FA pool representation discussed with these tables and as a result of antisense or deletion manipulation of fabB/F and fabH homologs are being biased by one member of the authors FA chain length summaries ie 16:1 and !8:1 with little change in other forms. 

We appreciate the reviewer’s suggestion! According to the reviewer’s suggestion, we have added the discussion about the variation of other forms of fatty acids. These revisions are in the paragraph 3 and 4 of section 2.3 and the paragraph 1and 2 of section 2.4.

In terms of effects of cerulenin on relative transcript pool sizes (can’t say “expression” as no growth rate data provided).   The authors on line 164 state: “indicating that the influence of cerulenin on the expression of fabH was weak” is totally off the mark.  fabH abundance in Fig1d is less than ¼ the level in control cultures, being more equivalent to d16 levels in untreated cultures.  Assuming that experimental cultures were started from older stock cultures (e.g. day 16), fabH expression only decreases or is diluted from this level during cerulenin (i.e. fabH expression is inhibited by this treatment).

Thanks for the comments! We completely agree with the reviewer. The expression level of fabH gene did not change significantly during the whole cultivation period under 7.5g/L cerulenin concentration (Figure 1d), which indicated that cerulenin also have specific inhibition on KAS I and KAS II in Synechococcus sp. PCC 7942. However, high cerulenin concentration could inhibit the growth of Synechococcus sp. PCC 7942 (Figure S1, as indicated in the above 4th point). Therefore, although there was no significant change during the whole cultivation period, the fabH expression level at 7.5 g/L cerulenin was much lower (Figure 1d) compared with that in the absence of cerulenin (Figure1b). This is probably caused by the growth inhibition at 7.5 g/L cerulenin concentration (Figure S1). So, the dynamic expressions of two related genes in Synechococcus sp. PCC 7942 may be the result of both growth and gene inhibition. This information was added to the paragraph 4 of section 2.2.

In terms of antisense and deletion mutant experiments in tables 3 and 4 the reader has no clue as to what growth stage (log, stationary, senescent cultures) are being compared.  Also their own indication on line 202 that “antisense fragment could inhibit the expression level of fabB/F” should be clearly demonstrated by qPCR comparisons of transcript abundance for all of your transformed lines with wild type strain.  This should easily be supplied as supplemental information. 

Thanks for your kind suggestion! The antisense and deletion mutant experiments in Tables 3 and 4 were performed in day 16 (stationary stage). We have added qPCR comparisons of transcript abundance for all of the transformed lines with wild type strain. This information was supplied as supplemental data (Figure S2).

The antisense fragment could inhibit the expression level of fabB/F and fabH.

As the following Figure S2 shown, the expression levels of fabB/F and fabH in antisense expression of fabB/F and fabH strains were down-regulated as compared with that in wild type strain during the whole growth period. The information was added to the paragraph 3 of section 2.3.

Please see Figure S2.

The review and reader would benefit from some discussion of why the fabB/F and particularly fabH knockouts in Synechoccocus are not lethal? Were are they getting the FA to continue growth from, certainly not BG11.  If fabH regulates entry into FA elongation cycle then lack of fabH should dramatical knock down at least total FA accumulation relative to cyanobacterial biomass…..  Again it should be simple to provide data on integration of your knockout cassette into the genome….. if its not in the correct place (which would be consistent with the data presented), then its not a knockout mutation…

Thanks for the comments! In the study, the deletion mutants were further confirmed by DNA sequence analysis. The sequencing data was provided as supplemental data (Figure S3).

Previous studies showed that fabH was a rate-limiting step and also the first condensation step in the fatty acid synthesis circle in plants and bacteria [Dehesh, et al., 2001; Brück, et al., 1996]. González-mellado et al. [2010] found that KAS III in sunflower had a new substrate specificity, which could participate in the synthesis of C6 to C10 fatty acids. Our present results showed that the C18 fatty acid decreased with the increase of C14 fatty acid when the gene fabH was deleted, suggesting that the Synpccw7942_1455 (fabH) in Synechococcus sp. PCC 7942 may only participate in the carbon chain extension (C<18) (not in the first condensation step in the fatty acid synthesis circle or regulate entry into FA elongation cycle) and had acyl-ACP carbon chain substrate specificity. We have revised Figure 3. It was also reported that some enzyme had the activity of KAS III, with the domain similar to FabH, but whose molecular weight is larger than FabH, initializing fatty acid synthesis in Pseudomonas aeruginosa [Leonard, et al., 1998].Therefore, it is assumed that fab(B/F)-and fabH- mutants were considered as strains deficient in fabB/F and fabH gene expression but not completely inactivation, or other proteins with similar activity executed corresponding functions (KAS III). These probably are the reason why the fabB/F and fabH knockouts in Synechococcus sp. PCC 7942 are not lethal.

The above information was added to the paragraph 4 of section 2.3, the paragraph 3 of section 2.4 and section 3.4

References:

Dehesh, K.; Tai, H.; Edwards, P.; Byrne, J.; Jaworski, J.G. Overexpression of β-ketoacyl-acyl-carrier protein synthase IIIs in plants reduces the rate of lipid synthesis. Plant Physiol. 2001, 125(2): 1103-1114.

Brück, F.M.; Brummel, M.; Schuch, R.; Spener, F. In-vitro evidence for feed-back regulation of beta-ketoacyl-acyl carrier protein synthase III in medium-chain fatty acid biosynthesis. Planta. 1996, 198(2): 271-278.

González-Mellado, D.; von Wettstein-Knowles, P.; Garcés, R.; Martínez-Force, E. The role of β-ketoacyl-acyl carrier protein synthase III in the condensation steps of fatty acid biosynthesis in sunflower. Planta. 2010, 231(6): 1277-1289.

Leonard, J.; Knapp, S.J.; Slabaugh, M.B.A Cuphea β-ketoacyl ACP synthase shifts the synthesis of fatty acids towards shorter chains in Arabidopsis seeds expressing Cuphea FatB thioesterases. Plant J. 1998, 13(5): 621–628.

Ln 254 and beyond, use complementation rather than “heter-complementation”

Thanks for the kind suggestion! We have modified “heter-complementation” into “complementation”. (Abstract) (Section 2.5) (Table 5)

Reviewer 2 Report

the article Short chain fatty acids biosynthesis relys on β-ketoacyl ACP synthase in Synechococcus sp. PCC need to be improved in adding more relevant background and literature in the discussion. The data presentation can be improved.

Authors’ responses:

the article Short chain fatty acids biosynthesis relys on β-ketoacyl ACP synthase in Synechococcus sp. PCC need to be improved in adding more relevant background and literature in the discussion. The data presentation can be improved.

Thanks for the comments! According to the reviewer’s suggestion, more relevant background and literature in the discussion were added. We also improved the data presentation. We highlight in yellow the changes made in revised version.

Reviewer 3 Report

Algae growth systems show great potential for the production of selected bioproducts. The main reasons for this are due to their growth rates and high concentrations of valuable biocompounds. Despite increasing interest in algae as multi-use feedstock,  there are many knowledge gaps that need to be filled in order to better understand their potential usage for biochemical production. In general, additional experiments are required to strengthen the Author’s conclusions. At current form, the manuscript lacks in some of its parts the sufficient scientific level for acceptance in the Marine Drugs journal. As outlined in the major points, the manuscript includes too preliminary data that adding very little new to the general knowledge of the possible usage of algae in the industrial-oil production.

Major points:

1.      In general the manuscript is badly organized. There is a lack of logical connection between the particular sections of the work.

2.      The style of writing and presenting of particular sections of the manuscript are words apart; example: in the results section there are many language and substantive mistakes (e.g. different kinds of lipids; according to the above results; to understand the changes of fatty acids, we studied the dynamic changes of fatty acids; conventional fatty acid composition,  etc.), many sentences are incoherent, while the introduction section indicates that Authors have some practical knowledge in the field of research.

3.      The manuscript should be checked by a native speaker. Many lexical borrowings caused that some sentences are illegible.

4.      The Materials and Methods section does not provide sufficient technical information to allow the experiments to be reproduced.  For instance it is unclear how the selection of Synechococcus sp. PCC7942 was done; what does ‘normal growth condition” mean?

5.      Authors should find and investigate the limiting steps in whole photosynthetic CO2 fixation process to increase lipid productivity in applied model.

6.      The manuscript is lacking of any proper statistical analyses

7.      The discussion section actually did not exist in the manuscript. The discussion section should be restricted to interpretation of the results.

8.      What is the industrial application of the obtained results?

Authors’ responses:

Algae growth systems show great potential for the production of selected bioproducts. The main reasons for this are due to their growth rates and high concentrations of valuable biocompounds. Despite increasing interest in algae as multi-use feedstock,  there are many knowledge gaps that need to be filled in order to better understand their potential usage for biochemical production. In general, additional experiments are required to strengthen the Author’s conclusions. At current form, the manuscript lacks in some of its parts the sufficient scientific level for acceptance in the Marine Drugs journal. As outlined in the major points, the manuscript includes too preliminary data that adding very little new to the general knowledge of the possible usage of algae in the industrial-oil production.

Thanks for the comments! According to the reviewer’s suggestion, we have added more background knowledge and discussion to section “Introduction” (paragraph 2,3,4) and section “Results and discussion”. Besides, additional experiment was conducted to strengthen our conclusions (section 2.3, paragraph 3). We highlight in yellow the changes made in revised version.

Major points:

1.      In general the manuscript is badly organized. There is a lack of logical connection between the particular sections of the work.

Thanks for the comments! According to the reviewer’s suggestion, we have reorganized the manuscript, strengthened the logical connection between each parts of the work.

2.      The style of writing and presenting of particular sections of the manuscript are words apart; example: in the results section there are many language and substantive mistakes (e.g. different kinds of lipids; according to the above results; to understand the changes of fatty acids, we studied the dynamic changes of fatty acids; conventional fatty acid composition,  etc.), many sentences are incoherent, while the introduction section indicates that Authors have some practical knowledge in the field of research.

Thanks for the comments! We carefully take the reviewer’s kind suggestion and revised our manuscript, including language and substantive mistakes, logic.

3.      The manuscript should be checked by a native speaker. Many lexical borrowings caused that some sentences are illegible.

Thanks for the comments! According to the reviewer’s suggestion, we have improved the English use in the manuscript.

4.      The Materials and Methods section does not provide sufficient technical information to allow the experiments to be reproduced.  For instance it is unclear how the selection of Synechococcus sp. PCC7942 was done; what does ‘normal growth condition” mean?

Thanks for the comments! We have added the sufficient technical information to the Materials and Methods section.

The reason for selection ofSynechococcus sp. PCC7942 was as follows: Although the fatty acids composition of microalgae varied with species and strains, but some cyanobacteria had strong ability to synthesize short carbon chain fatty acids. A strain of filamentous cyanobacteria Trichodesmium erythraeum has been found to produce 27% to 50% C10 fatty acids. Karatay et al. reported Synechococcus sp. could accumulate about 23.8% of caprylic acid (C10:0) and myristic acid (C14:0) under nitrogen deficiency condition. Therefore, cyanobacteria may be an ideal species for the production of short carbon fatty acids. In addition, cyanobacteria, as prokaryotes, is more suitable for gene manipulation than eukaryotic microalgae, and some of which have completed the whole genome sequencing. Thus, Synechococcussp. PCC 7942, whose genome have been sequenced, was selected to explore the carbon chain fatty acid synthesis in our study.

Cell growth in BG11 medium was set as normal growth condition. Different cerulenin concentrations were set as follows: 0g/L, 1g/L, 2.5g/L, 5g/L and 7.5g/L. Stock solution of cerulenin (100mg/L, preserved in ethyl alcohol, -20) was added to BG11 medium with the final concentrations at 0 day.

Some of the above information was added to section “Introduction” (last paragraph) and section “Material and methods” (3.1).

5.      Authors should find and investigate the limiting steps in whole photosynthetic CO2 fixation process to increase lipid productivity in applied model.

Thanks for the comments! This is a very valuable suggestion. The previous researches showed that fatty acids synthetic system started from acetyl-CoA carboxylase (ACCase), catalyzing the conversion of acyl-CoA to malonyl CoA, which step was thought to be a limiting step. KASIII had also been proved a limiting step of fatty acid biosynthesis in cyanobacteria Synechococcussp. PCC7002. The above two limiting steps might be the best options to increase lipid productivity in the whole photosynthetic CO2 fixation process. Overexpression of ACCase and KASIII genes had succeeded to regulate the lipid synthesis. These revisions were added to section 2.6 (last paragraph) and Figure 3.

6.      The manuscript is lacking of any proper statistical analyses

We appreciate the reviewer’s suggestion. We have added proper statistical analyses to the results (paragraph 4 of section 2.1, paragraph 3 and 4 of section 2.3), Figures and Tables.

7.      The discussion section actually did not exist in the manuscript. The discussion section should be restricted to interpretation of the results.

Thanks for the comments! According to the instructions for authors, interpretation of the results and discussion section can be merged together. Therefore, the results and discussion were merged in the manuscript. We have also modified our discussion as suggested.

8.      What is the industrial application of the obtained results?

Thanks for your questions! Short chain fatty acids are valued as a high-added functional material in cosmetics for its extraordinary characteristics. The results may provide scientific guidance for the future of developing short chain fatty acid resources in microalgae by means of metabolic engineering and molecular biology. In our study, the synthetic path way of short chain fatty acids in microalgae was proposed. We expected to use metabolic engineering or molecular biological technologies to regulate microalgae SCFA accumulation or construct engineering microalgae for SFCAs-derived cosmetics in the future.